# General Practitioners’ Own Traumatic Experiences and Their Skills in Addressing Patients’ Past History of Adversities: A Cross-Sectional Study in Portugal

**DOI:** 10.3390/healthcare11172450

**Published:** 2023-09-01

**Authors:** Mariana Sá, Paulo Almeida Pereira, Ivone Castro-Vale

**Affiliations:** 1Medical Psychology Unit, Department of Clinical Neurosciences and Mental Health, Faculty of Medicine, University of Porto, 4200-319 Porto, Portugal; ivonecastrovale@med.up.pt; 2Unidade de Saúde Familiar Famílias, Agrupamento de Centros de Saúde Entre Douro e Vouga I—Feira/Arouca, 4535-086 Lourosa, Portugal; 3Institute of Management and Health Organizations, Universidade Católica Portuguesa, 3504-505 Viseu, Portugal; ppereira@ucp.pt; 4i3S-Institute of Research and Innovation in Health, University of Porto, 4200-135 Porto, Portugal

**Keywords:** adverse childhood experiences, traumatic events, clinical communication skills, empathy, confidence, general practitioners

## Abstract

Addressing trauma has been found to be important for primary care patients, as it can improve their health-related outcomes. We aimed to assess how Portuguese general practitioners’ (GPs) past history of traumatic events (TEs) and adverse childhood experiences (ACEs) influence their clinical communication skills when addressing their patients’ past history of adversities. An online survey was circulated by email to GPs’ associations and through GPs’ social media groups. A sample of 143 GPs participated in this study. GPs’ exposure to ACEs and TEs was assessed using the Childhood Trauma Questionnaire-Short Form and the Life Events Checklist for the DSM-5. To evaluate clinical communication skills, we adapted the Self-confidence Scale and used the Jefferson Scale of Physician Empathy. GPs identified barriers to addressing trauma routinely, including a lack of time (86.7%) and a fear of causing further patient suffering (56.6%). GPs’ exposure to TEs and ACEs was positively correlated with scores in some dimensions of self-confidence and empathy (*r* values varying from 0.170 to 0.247). GPs exposed to traumatic experiences felt more confident when addressing their patients’ adversities and were more empathic when conducting therapeutic relationships. This study shows that GPs with a history of traumatic experiences are able to address their patients’ adversities; however, they lack proper training and better patient care conditions, such as more time and more resources available for patient guidance.

## 1. Introduction

Adverse childhood experiences (ACEs) refer to direct or indirect harm caused to children, primarily resulting from abuse, neglect, and tumultuous domestic environments [1]. ACEs include childhood maltreatment, such as emotional abuse, physical abuse, sexual abuse, physical neglect, emotional neglect, and bullying. In addition, these experiences can also be related to those aspects of children’s environments that can undermine their sense of safety, stability, and bonding, such as growing up in a household with drug abuse and mental health problems, witnessing domestic violence, parental separation, or household members being in prison [1,2]. According to the World Health Organization, the prevalence of childhood maltreatment varies from country to country and is often underestimated [3]. A systematic review estimated that the prevalence of ACEs in school-aged youth in the United States of America ranges from 41% to 97% [4]. Data from European studies suggested that 42% of adults had suffered at least one ACE whilst growing up, with 19% having suffered more than one ACE [5]. Dias et al. (2015) concluded that childhood maltreatment is a significant problem in Portugal, and found that around 14.7% of Portuguese adults self reported moderate or severe childhood maltreatment exposure, with 67% reporting exposure to more than one form [6]. 

Since ACEs are often insidious and vary in severity, they can directly and indirectly disrupt the child’s physical or mental health development and can result in serious consequences in adulthood [6,7]. ACEs increase the risk of substance abuse and engagement in other high-risk behaviors and are also associated with the development of mental health disorders and chronic medical conditions [1,6,8,9,10]. 

Although some ACEs include traumatic events (TEs), they are not restricted to them. TEs are very stressful and traumatic experiences that can occur throughout the lifetime, and have been typified in the latest versions of the American Medical Association Diagnostic and Statistical Manual of Mental Disorders (DSM) criteria for posttraumatic stress disorder (PTSD). TEs are frequently experienced in the population, with a lifetime prevalence ranging from 64% to 90%, and they are also known to be important contributors to an increased risk of mental and physical morbidity [11,12,13,14]. According to the fifth edition of the DSM (DSM-5), the experience of a TE occurs when a person is directly exposed to actual or threatened death, serious injury, or sexual violence, or witnesses such an event, or when they learn that such an event has happened to someone who is close to them, such as a family member or a close friend [15]. TEs are associated with a high prevalence of chronic diseases, due to the dysregulation of the biological stress response systems, such PTSD, depression, and obesity, among others [12,13,14,15,16].

Studies have shown that the early detection of both ACEs and TEs, as well as a timely intervention at this level, can decrease the risk of problems associated with trauma exposure [17]. As primary care is generally the first point of healthcare contact for most people, GPs are in a privileged position to explore their patients’ exposure to ACEs and TEs. Accordingly, one of GPs’ core competencies is the ability to adopt a holistic approach and provide longitudinal care. This involves approaching patients’ health problems by considering all relevant dimensions, i.e., physical, psychological, social, cultural, and existential factors, which includes issues related to identity, death, futility, spirituality, and religion [18,19]. 

Evidence has shown that patients appreciate when their GP routinely addresses their history of traumatic experiences, and that empathic communication influences patient satisfaction, helps to strengthen the doctor–patient relationship, and can also predict physiological, behavioral, and emotional outcomes [20,21,22].

Existing research shows limited focus on GPs’ perspective of the importance of addressing patients’ past traumatic experiences. Moreover, the methods GPs employ to communicate about these experiences in clinical practice remain underexplored. However, barriers have been identified which make the approach to patients’ traumatic experiences more difficult, such as a lack of training in clinical communication, a lack of time, and fear of harming the patient and of damaging the doctor–patient relationship [23,24]. Accordingly, patients’ history of exposure to ACEs and TEs is still under-recognized in primary healthcare [8,14,25].

In addition to the barriers that GPs face, raising the topic of trauma can also be a challenging task for them, depending on their own personal experiences, which can deeply influence their perceptions, roles, and medical practice. Furthermore, the practice of medicine is a physically and emotionally challenging profession, which itself constitutes a risk factor for numerous traumatic experiences. Therefore, GPs’ own history of ACEs and TEs is also an important aspect to consider. To our knowledge, no studies exist that focus on the prevalence of ACEs and TEs among GPs from Portugal, or indeed from any other country. 

The experience of ACEs or TEs by physicians themselves can result in: (1) positive psychological changes, as the physician is likely to have developed more coping mechanisms to deal with adverse experiences [26], or (2) mental distress, including posttraumatic stress symptoms, such as re-experiencing, the avoidance of reminders of past events, negative cognitions and mood, and hyperarousal, among others [14,15,16]. These symptoms can negatively influence the way that GPs communicate, especially when it comes to exploring their patients’ traumatic experiences. To the best of our knowledge, there are no studies focusing on these specific associations, especially with regard to primary care providers. 

The purpose of this research was to assess Portuguese GPs’ perceptions, practices, and clinical communication skills when addressing their patients’ traumatic experiences. We also intended to establish associations between the way GPs communicate and their personal history of adverse experiences. 

Accordingly, several hypotheses were formulated, namely: (1) GPs will encounter barriers that have already been identified in other studies which make it more difficult to routinely address patients’ traumatic experiences [11,23,27]; and (2) GPs who have experienced TEs or ACEs tend to address these topics more easily, as they may be less likely to underestimate the realities of trauma and may feel more confident engaging in sensitive communications due to their shared experiences. 

In summation, we believe that the study of the associations between GPs’ own traumatic experiences and their clinical communication skills in addressing their patients’ history of adversities can add valuables reflections and present new questions to the extant literature in the field. These correlations might well help provide better communication practices and understanding between GPs and their patients, which in turn is beneficial to the therapeutic process. Overall, this study can draw attention to the need for continuous clinical communication training and better conditions in which to practice.

## 2. Materials and Methods

### 2.1. Participants and General Procedures

GP specialists and residents practicing in Portugal were invited via an email sent from GPs’ associations and through GPs’ social media groups to participate in this study on a voluntary basis. The survey’s dissemination did not follow any specific criteria, as instead, it targeted all those GPs registered in the above-mentioned associations and those who were active in the social media groups. No randomization was performed. The respondents were asked to complete a confidential online self-reporting survey which was created using Google Forms^®^. 

In this cross-sectional study, all the participants were informed about the aims of the study and were asked to give their free and informed consent to participate. Only those who completed and submitted the survey were counted as participants. A sample of 143 participants answered the online survey from 2 June to 11 July 2022. The estimated number of GPs in Portugal in 2022 was 9000 [28]. An effort was made to reach as many GPs residents and specialists as possible. For the sample size of 143 and for expected correlations of 0.25, the power for the analysis was 85.8%.

The coding and recording of the obtained data was organized in a computer database, which was only accessible to the research team and was created on the Google Sheets^®^ platform. In addition to restricting access, we took other measures in order to ensure participants’ privacy and confidentiality, such as using a password and regularly monitoring account activity for suspicious or unauthorized access.

The study was approved by the Ethics Committee of our university (Comissão de Ética para a Saúde do Centro Hospitalar São João/Faculdade de Medicina da Universidade do Porto, Approval number: 60-22).

### 2.2. Instruments 

Sociodemographic data and professional characterization of the participants (e.g., whether they were from the public or the private sector and also the type of healthcare unit) were obtained. 

The Portuguese version of the Childhood Trauma Questionnaire—-Short Form (CTQ-SF) was used to assess GPs’ exposure to childhood adversities [29,30]. The CTQ-SF is a 28-item self-reporting measure, which has been validated in a Portuguese non-clinical sample (Cronbach’s alpha ranged from 0.84, total score, to 0.47, physical neglect). Each item is rated on a 5-point Likert-type scale which evaluates the frequency of the events, ranging from 1 (never) to 5 (always). The CTQ-SF measures five different types of maltreatment experienced during childhood and adolescence, namely: emotional abuse; physical abuse; sexual abuse; emotional neglect; and physical neglect. In this study, we used each scale, and also the total CTQ-SF score. Higher scores correspond to higher maltreatment during childhood and adolescence. 

The Portuguese version of the Life Events Checklist for DSM-5 (LEC-5, Portuguese version Ferreira, Ribeiro, Santos & Maia. Unpublished manuscript, Department of Psychology, University of Minho, Braga, Portugal) was used to assess GPs’ exposure to TEs, as defined by DSM-5 criterion A for PTSD [31]. LEC-5 consists of 17 items used to assess 16 TEs, as well as item 17, which requests the identification of any other very stressful event that was not previously included. Next, participants indicate varying levels of exposure for each item, using the six nominal categories of responses, namely: (1) happened to me; (2) witnessed it; (3) learned about it; (4) part of my job; (5) not sure; or (6) doesn’t apply. Participants may endorse multiple levels of exposure for the same type of TE. A positive trauma endorsement was indicated when individuals selected either of the first four response options [32]. In our analysis, TEs types were grouped into three clusters, in accordance with the types previously described by Contractor, Weiss, Natesan and Elhai [32], i.e.,: accidental/injury TEs; victimization TEs; and predominant death threat TEs. For our study, following the Contractor et al. procedures [32], the analysis of item 17 was not considered because of the ambiguity of the obtained content. In our sample, the value of Cronbach’s alpha was 0.90 for the total traumatic load, 0.83 for the predominant death threat TEs cluster, 0.70 for the victimization TEs cluster, and 0.77 for the accidental/injury TEs cluster. 

To assess GPs’ clinical communication perceptions and practices when addressing patients’ adversities, we formulated eight questions based on our clinical experience (see Appendix A). In addition, two further questionnaires were used to evaluate GPs’ clinical communication skills, namely self-confidence and empathy when dealing with patients’ traumatic experiences.

To assess GPs’ confidence when addressing their patients’ past traumatic experiences, we adapted the Self-confidence Scale for clinical communication skills [33,34]. The original scale has 17 items; however, in accordance with the aims of our study, we only used 11 items from a Likert-type scale, scoring from 1 (no confidence), to 7 (all confidence). The scale focused on confidence to address traumatic experiences. As the scale used in our study was an adaptation of the original scale, exploratory factor analysis was used to determine the dimensions for the 11 items (see Appendix A). The saturation of the variables in each factor was always above the required minimum of 40%. Table 1 presents the items of each dimension of the adapted Self-confidence Scale obtained using the exploratory factor analysis. For each dimension, the values were determined by calculating the mean of the answers to the items that constitute them. Cronbach’s alpha was 0.90 for the total score, 0.86 for the relationship building dimension, 0.84 for the directive facilitation dimension, and 0.76 for the nondirective facilitation dimension. The Portuguese version of the survey used in our study is presented in Appendix A. 

The Portuguese version of the Jefferson Scale of Physician Empathy (JSPE-VP) was used to assess GPs’ empathy in therapeutic relationships [35,36]. JSPE-VP is a 20-item Likert-type self-report survey answered on a 7-point scale (1 = strongly disagree, 7 = strongly agree) and it consists of three dimensions: perspective taking; compassion; and standing in the patient’s shoes. In each dimension, the higher scores correspond to higher levels of empathy. 

### 2.3. Statistical Analysis 

Data analysis was performed using SPSS^®^ (Version 27.0) [37]. The normal distribution was verified using the Kolmogorov–Smirnov test, and thus only the parametric tests were used. Descriptive statistics were presented as frequencies (*n*), percentages (*%*), mean (*M*), standard deviation (*SD*), minimum (*Min.*), and maximum (*Max.*) values for the quantitative variables. The Cronbach’s alpha was used as a measure of internal consistency, with a coefficient of internal consistency of 0.80 or more being regarded as adequate, and values from 0.60 to 0.80 being acceptable, albeit considered to be weaker. Statistical tests were used to determine whether the associations between variables observed in our sample were statistically significant and whether the findings could be inferred for the population. The alpha level of 5% was adopted as a reference, meaning that inference occurs with a probability of error less than 5%. Exploratory factor analysis was used to extract factors for the adapted Self-confidence Scale used in our survey. The analysis of variance (ANOVA) parametric test was used to study the relationship between years of clinical experience and the adapted Self-confidence Scale. To study the association between quantitative variables, the Pearson correlation coefficient was used as a measure of linear association. No adjustments were made for multiple comparisons, and there were no missing data for the variables under study.

## 3. Results

Table 2 presents the characteristics of the sample of GPs studied. Descriptive analysis for CTQ-SF, LEC-5, the adapted Self-confidence Scale, and JSPE-VP are presented in Table 3. Differences between genders for these variables were only significant for the JSPE-VP scale, with higher scores for the female gender. The frequencies related to LEC-5 and the distribution of the multiple answers for each trauma type along the type of exposure can be found in Appendix A.

### 3.1. GPs’ Clinical Communication Skills 

Regarding GPs’ clinical communication perceptions and practices when addressing patients’ past history of exposure to TEs or ACEs, most participants failed to routinely ask patients about their traumatic experiences (89.5%). Indeed, they only explored these issues in some specific contexts (83.9%), the most frequent being when they were attending patients with depressive and/or anxious symptoms (97.5%). In our sample, 58.7% of the participants considered that it is important to address patients’ traumatic experiences, and 36.4% considered that this is very important to them. However, only 25.2% of the GPs agreed that patients would like these issues to be routinely explored by their GP, and 4.2% totally agreed. The main constraints pointed out by participants were a lack of consultation time (86.7%); few available resources for subsequent patient guidance (63.6%); fear of causing more harm or suffering to the patient (56.6%); and a lack of training in clinical communication (54.5%). Regarding the reasons for patients not mentioning their traumatic experiences during their consultation, most participants pointed out the following as being the perceived reason: patients’ fear of not being understood (79.0%); shame (78.3%); and the suffering associated with remembering the events (72.0%).

### 3.2. GPs’ Clinical Communication Skills’ Association with Years of Clinical Experience and Previous Training in Clinical Communication

In our sample of GPs, we found evidence of an association between years of clinical experience and self-confidence in addressing patients’ traumatic experiences (Table 4). 

The mean for the adapted Self-Confidence Scale total score was higher for GPs with 5–10 years of clinical experience (*M* = 5.48) and for those with more than 10 years (*M* = 5.43) of clinical experience, but was lower for those with less than 2 years (*M* = 4.72). These differences were significant (*F*_(3.139__)_ = 3.824; *p* = 0.011). The mean for the dimension relationship building was higher for GPs with 5–10 years of clinical experience (*M* = 5.75), followed by those with more than 10 years (*M* = 5.48), and lower for GPs with less than 2 years (*M* = 5.03), and the differences are significant (*F*_(3.139__)_ = 2.838; *p* = 0.040). The mean for the dimension directive facilitation was higher for GPs with more than 10 years of clinical experience (*M* = 5.22), followed by those with 5–10 years (*M* = 4.75), and lower for GPs with less than 2 years (*M* = 3.83), and the differences were significant (*F*_(3.139)_ = 8.010; *p* < 0.001). Accordingly, it can be seen that the global scale and these dimensions tended to increase with increasing clinical experience. No significant associations were found for the nondirective facilitation dimension.

No significant association was found between previous training in clinical communication and GPs’ self-confidence. With regards GPs’ empathy, no significant associations were found between JSPE-VP scores and years of clinical experience and previous training in clinical communication.

### 3.3. The Association between GPs’ Past History of Traumatic Experiences and Clinical Communication Skills 

In our sample of GPs, we found positive significant correlations between the CTQ-SF dimensions physical abuse (*r* = 0.171, *p* = 0.042), and physical neglect (*r* = 0.199, *p* = 0.018), and the directive facilitation dimension of the adapted Self-Confidence Scale. We also found two positive significant correlations between TEs measured with LEC-5 (traumatic load, *r* = 0.185, *p* = 0.028; predominant death threat traumas, *r* = 0.208, *p* = 0.013) and the relationship building dimension of the adapted Self-Confidence Scale. Victimization traumas and relationship building dimension were associated at a trend level (*r* = 0.165, *p* = 0.050) and none of the other associations were statistically significant (Table 5). These correlations were controlled for gender. 

Table 6 shows the associations between GPs’ traumatic experiences and empathy results. We found a positive significant correlation between childhood physical abuse and the JSPE-VP’s perspective-taking dimension (*r* = 0.196, *p* = 0.020) and also between childhood emotional abuse and the JSPE-VP’s standing in patient’s shoes dimension (*r* = 0.170, *p* = 0.043). Eight significant positive correlations were found between LEC-5 and JSPE-VP, namely: the traumatic load and JSPE-VP’s global score (*r* = 0.185; *p* = 0.028), and the standing in patient’s shoes dimensions (*r* = 0.221; *p* = 0.008); the victimization traumas cluster and the JSPE-VP’s global score (*r* = 0.242; *p* = 0.004), perspective taking (*r* = 0.243; *p* = 0.004), and the standing in patient’s shoes dimensions (*r* = 0.212; *p* = 0.011); as well as the predominant death threat traumas cluster and the JSPE-VP’s global score (*r* = 0.212; *p* = 0.011), perspective taking dimension (*r* = 0.177; *p* = 0.035), and the standing in patient’s shoes dimension (*r* = 0.247; *p* = 0.003). These correlations were also controlled for gender. 

## 4. Discussion

This study aimed to assess GPs’ clinical communication practices, perceptions and skills when addressing their patients’ history of TEs and ACEs, as well as the influence of their own adverse experiences on these skills. To our knowledge, this is the first study to focus on these associations. 

According to our results, most GPs failed to routinely ask their patients about their past traumatic experiences, with more than 50% of the GPs citing the following principle reasons: a lack of time, few resources available for subsequent guidance, fear of causing more harm or suffering to the patient, and a lack of training in clinical communication. These results are in accordance with previous studies, which were also conducted in primary healthcare services and focused on GPs’ difficulties promoting trauma screening and on GPs’ experiences with patients with PTSD [23,24,38,39].

A recent study showed that patients with more psychiatric issues and those who have greater trust in their GP are the ones most likely to be screened for trauma and PTSD [39]. Accordingly, in our study, 97.5% GPs answered “In depressive and/or anxious symptoms” when asked about the specific contexts in which they address their patients’ traumatic experiences. 

Regarding the perceived reasons why patients fail to mention their trauma issues during the consultation, more than half of our sample of GPs highlighted the following reasons: fear of not being understood, shame, not wanting to suffer when remembering the events, or not wanting to bother the GP with these matters. Interestingly, studies that have addressed these issues from the patients’ perspective have shown that patients like their GP to routinely address trauma exposure, and that they often feed their physicians with small hints over time to test their openness to hearing about their life experiences and their ability to respond appropriately [20,21]. On the other hand, some patients do not believe that it is the GP’s role to explore these topics and have fears of “appearing crazy” [38]. 

Our study found a significant positive association between years of clinical experience and GPs’ confidence communicating when exploring their patients’ past history of traumatic experiences. According to some studies, the longitudinal care and consultation experiences were found to be the principle reasons that contribute to the promotion of a good doctor–patient relationship [40]. Based on our experience, we consider that at the primary care level, cumulative appointments over the years strengthen the doctor–patient relationship. Accordingly, an increase in years of clinical experience could contribute to an increase in physicians’ levels of confidence, as this relationship becomes stronger and deeper. Indeed, evidence shows that trauma screening appears to be more frequent over the course of relationship building [39]. 

On the other hand, no significant associations were found between empathy and years of clinical experience. Empathy is known to be predominantly cognitive, rather than affective or emotional [41], and it consists of the ability to cognitively understand patients’ perspectives, feelings and experiences, combined with the capacity to communicate this understanding back to the patient, and the intention to help them [41,42]. Some studies have shown that empathy levels decrease with increasing years of experience, long working hours, an increase in stress, and lack of sleep [43,44]. Thus, empathy appears to be more associated with physicians’ well-being and reduced fatigue symptoms, rather than with an increase in their clinical experience. The non-significant associations between previous training in clinical communication and empathy, and self-confidence, could be explained by the reduced number of participants who have postgraduate training in clinical communication in our sample of GPs (*n* = 20). 

According to our initial hypothesis, our study showed that the experience of childhood adversities or TEs by GPs was positively correlated with scores in certain dimensions of the adapted Self-Confidence Scale and JSPE-VP. As stated before, to our knowledge, there are no existing studies that focus on these specific associations. However, there is also some evidence that individuals can mature psychologically as a result of trauma or stressful events, resulting in post-traumatic growth [26,45]. This concept was created by Calhoun and Tedeschi and is related to positive psychological changes after exposure to TEs [26]. We consider that this concept could make GPs more empathetic, as it may become easier for them to understand patients’ perspectives and experiences if they themselves have also been exposed to trauma. Similarly, GPs could thus become more confident, as they are able to predict patients’ feelings or emotions more easily and can anticipate how to deal with them. However, on the other hand, physicians who have been exposed to traumatic adversities can also develop or enhance active coping mechanisms that help them to manage the negative effects of trauma exposure. Developing this hypothesis, this study should be replicated in other countries to assess how different biopsychosocial contexts change the effects of GPs’ traumatic experiences on their empathy and level of confidence to deal with their patients’ past traumatic experiences.

The associations between GPs’ own traumatic experiences and their clinical communication skills in addressing patients’ past history of adversities were controlled for gender, which is a strong point of this pioneer study. In our sample of GPs, female gender represented 76.9% of the total sample size. Furthermore, evidence has shown that women are generally more empathetic than men, and that female GPs and medical students often obtain significantly higher scores for empathy when compared with their male counterparts [36,44,46]. 

Another strength of our research was the use of validated instruments to evaluate GPs’ adverse experiences and communication skills when addressing their patients’ traumatic issues. The use of an online survey contributes to attaining a greater geographic reach, enabling participants to fill it out at any time and in any place, without any influence from the researchers. 

In our study, we did not use any measure of GPs’ psychopathology. Therefore, it would be interesting in futures studies to assess whether GPs who have been exposed to TEs or ACEs developed PTSD symptoms, depression, or other mental health disorders that could influence their communication when addressing their patients’ traumatic experiences, or if, on the contrary, they develop posttraumatic growth

Another limitation of our study is related to the positive correlations with statistical significance we found between GPs’ adverse experiences and clinical communications skills, where the correlation values ranged from 0.170 to 0.247. These results indicate weak associations and could be explained by the small sample size of our study. 

We acknowledge that another limitation of our study is that our sample may not be representative of GPs overall, due to the small size of the sample and the recruitment method used. We failed to obtain responses from GPs from all of the regions of Portugal. Furthermore, the use of an online survey implies some inherent constraints, such as: it is more likely to be completed by more digitally literate physicians; the time required to complete the survey may limit its completion, as there is no commitment to the researcher; the questions can be interpreted in different ways by the participants without the possibility of seeking clarification from the researcher at the time of filling out the questionnaire. These limitations warrant future studies to confirm our results, which should include larger and representative samples and possibly other methodological designs, such as multicenter trials, including in other countries, and with the possibility of distributing and filling out the survey in person. Future studies should also include measures of post-traumatic growth and psychopathology in GPs. 

## 5. Conclusions

GPs who have been more exposed to adversities in childhood or TEs showed more confidence in assessing their patients’ traumatic experiences and were more empathetic in their therapeutic relationships. However, the approach to patients’ past history of trauma is affected by organizational constraints and by GPs’ fear of damaging the doctor–patient relationship. Efforts should be made at the organizational level of primary care services to promote traumatic experiences screening. Training in clinical communication should be provided regularly and specifically with GP trauma-informed care best practices, such as those proposed by Tomaz and Castro-Vale [25]. GPs should have better conditions to practice, such us more time in the consultation, and also a greater availability of resources to which they can refer patients if necessary, like psychologists, psychiatrists, and maybe having them work together, closer to primary care services. They should also be trained to know how to deal with strong emotions. These measures might improve GPs’ confidence, and also their empathic capacities, even more. GPs’ medical education should emphasize the importance of addressing traumatic experiences as a means for obtaining increased levels of patient satisfaction and positive health outcomes. 

In times of constant conflict, human trafficking, the increase in refugees, and child sexual abuse, it is even more important to invest in the care of people with a traumatic history and also in those who care for them.

## Figures and Tables

**Table 1 healthcare-11-02450-t001:** Items of each dimension of the adapted Self-confidence Scale, as obtained using the exploratory factor analysis.

Dimensions	Items
Relationship building	6. Identify the patient’s non-verbal communication. 7. Build a good clinical relationship with the patient. 8. Assertively deal with emotions. 9. Identify unexpressed feelings for the patient. 10. Recognize your own feelings towards the patient (negative or positive). 11. Deal with an anxious or depressed patient.
Directive facilitation	3. Shift the agenda from the patient’s to your own at the appropriate time.
	4. Maintain the flow of the interview. 5. Provide an adequate structure for the consultation.
Nondirective facilitation	1. Avoid interrupting the patient.
	2. Avoid making the patient feel rushed.

**Table 2 healthcare-11-02450-t002:** Sociodemographic and professional characterization of the total sample of GPs.

*n* = 143		*M*	*SD*
Age		40.0	13.2
		* **n** *	**%**
Gender	Male	33	23.1
	Female	110	76.9
Professional category	GP resident	60	42.0
	GP specialist	83	58.0
Years of clinical experience	<2 years	12	8.4
	2–5 years	47	32.9
	5–10 years	29	20.3
	>10 years	55	38.5
Place of work	FHU-A	38	26.6
	FHU-B	90	62.9
	PHCU	12	8.4
	Private	3	2.1
Geographical area of work	North	42	29.4
	Center	26	18.2
	Lisboa and Vale do Tejo	70	49.0
	Alentejo	5	3.5
Clinical Communication Training	None	83	58.0
	Pre-graduate training	40	28.0
	Postgraduate training	20	14.0

FHU-A: Model A Family Health Unit; FHU-B: Model B Family Health Unit; GP: general practitioner; *M*: mean; *n*: number; PHCU: Primary Health Care Unit; *SD*: standard deviation.

**Table 3 healthcare-11-02450-t003:** Descriptive analysis for CTQ-SF, LEC-5, adapted Self-confidence Scale, and JSPE-VP, by gender.

		Male (*n* = 33)		Female (*n* = 110)				Cohen
Instrument		*M*	*SD*	*M*	*SD*	*t*	*p*	*d*
**CTQ-SF**	CTQ-SF total score	37.64	9.61	36.86	9.22	0.418	0.677	0.083
	Emotional abuse	7.12	2.33	7.30	3.02	−0.313	0.755	−0.062
	Emotional neglect	7.94	3.22	7.50	2.81	0.761	0.448	0.151
	Sexual abuse	5.03	0.17	5.27	1.12	−1.240	0.217	−0.246
	Physical abuse	5.76	2.28	5.45	0.98	1.134	0.259	0.225
	Physical neglect	6.12	2.00	5.63	1.31	1.670	0.097	0.331
**LEC-5**	Traumatic load	7.09	5.39	6.15	4.67	0.974	0.332	0.193
	Accidental/injury traumas	2.88	1.87	2.42	1.76	1.302	0.195	0.258
	Victimization traumas	1.33	1.19	1.21	1.23	0.514	0.608	0.102
	Predominant death threat traumas	2.88	2.68	2.53	2.35	0.729	0.467	0.145
**Adapted** **Self-Confidence Scale**	Total score	5.37	0.89	5.26	0.79	0.715	0.476	0.142
	Relationship building	5.42	0.92	5.45	0.79	−0.198	0.843	−0.039
	Directive facilitation	5.07	1.09	4.68	1.08	1.802	0.074	0.358
	Nondirective facilitation	5.68	1.17	5.53	1.09	0.680	0.497	0.135
**JSPE-VP**	Global score	98.94	11.49	104.11	9.79	−2.553	* 0.012	−0.507
	Perspective taking	58.21	7.03	60.91	5.86	−2.210	* 0.029	−0.439
	Compassion	26.73	3.63	27.95	2.83	−2.042	* 0.043	−0.405
	Standing in patient’s shoes	14.00	3.48	15.25	3.06	−1.985	* 0.049	−0.394

CTQ-SF: Childhood Trauma Questionnaire-Short Form; JSPE-VP: Portuguese version of Jefferson Scale of Physician Empathy; LEC-5: Life Events Checklist for DSM-5; *M*: mean; *SD*: standard deviation; * *p* < 0.05; *d* > 0.20 small effect; *d* > 0.50 medium effect; *d* > 0.80 large effect (absolute values).

**Table 4 healthcare-11-02450-t004:** Associations between adapted Self-confidence Scale and years of clinical experience.

	*n* = 143	Years of Clinical Experience										
		<2 Years *n* = 12		2–5 Years *n* = 47		5–10 Years *n* = 29		>10 Years *n* = 55				
		*M*	*SD*	*M*	*SD*	*M*	*SD*	*M*	*SD*	*F* _(3.139)_	*p*	*Eta* ^2^
	Total Score	4.72	0.84	5.14	0.79	5.48	0.77	5.43	0.78	3.824	* 0.011	0.076
**Adapted** **Self-Confidence Scale**	Relationship building	5.03	0.92	5.32	0.83	5.75	0.63	5.48	0.84	2.838	* 0.040	0.058
	Directive facilitation	3.83	1.07	4.51	1.01	4.75	1.07	5.22	0.98	8.010	* 0.000	0.147
	Nondirective facilitation	5.13	1.25	5.52	1.08	5.76	1.24	5.60	1.03	0.966	0.411	0.020

*M*: mean; *n*: number; *SD*: standard deviation; *Eta^2^*: percentage of the variable explained; * *p* < 0.05.

**Table 5 healthcare-11-02450-t005:** Partial correlations between CTQ-SF and LEC-5, and the adapted Self-Confidence Scale, controlling for gender.

*n* = 143		Adapted Self-Confidence Scale			
		Total Score	Relationship Building	Directive Facilitation	Nondirective Facilitation
**CTQ-SF**					
CTQ-SF total score	Correlation (r) *p*	0.074	0.045	0.143	−0.011
		0.379	0.594	0.089	0.898
Emotional abuse	Correlation (r) *p*	0.075	0.082	0.076	0.006
		0.377	0.331	0.366	0.943
Emotional neglect	Correlation (r) *p*	0.071	0.045	0.158	−0.045
		0.399	0.591	0.061	0.597
Sexual abuse	Correlation (r) *p*	−0.103	−0.108	−0.064	−0.079
		0.224	0.202	0.448	0.352
Physical abuse	Correlation (r) *p*	0.059	−0.037	0.171	0.071
		0.484	0.663	* 0.042	0.402
Physical neglect	Correlation (r) *p*	0.144	0.087	0.199	0.095
		0.087	0.304	* 0.018	0.263
**LEC-5**					
Traumatic load	Correlation (r) *p*	0.115	0.185	0.093	−0.084
		0.173	* 0.028	0.269	0.317
Accidental/injury traumas	Correlation (r) *p*	0.054	0.105	0.047	−0.087
		0.525	0.213	0.578	0.305
Victimization traumas	Correlation (r) *p*	0.091	0.165	0.044	−0.067
		0.283	0.050	0.604	0.431
Predominant death threat traumas	Correlation (r) *p*	0.144	0.208	0.130	−0.071
		0.087	* 0.013	0.124	0.400

CTQ-SF: Childhood Trauma Questionnaire—Short Form; LEC-5: Life Events Checklist for DSM-5; *n*: number; * *p* < 0.05.

**Table 6 healthcare-11-02450-t006:** Partial correlations between CTQ-SF and LEC-5, and JSPE-VP, controlling for gender.

*n* = 143		JSPE-VP			
		Global Score	Perspective-Taking	Compassion	Standing in Patient’s Shoes
**CTQ-SF**					
CTQ-SF total score	Correlation (r) *p*	0.062	0.091	−0.098	0.119
		0.460	0.283	0.248	0.159
Emotional abuse	Correlation (r) *p*	0.086	0.103	−0.097	0.170
		0.311	0.224	0.250	* 0.043
Emotional neglect	Correlation (r) *p*	−0.007	0.016	−0.130	0.069
		0.930	0.850	0.124	0.414
Sexual abuse	Correlation (r) *p*	−0.009	0.039	−0.134	0.022
		0.912	0.643	0.112	0.795
Physical abuse	Correlation (r) *p*	0.158	0.196	0.110	0.026
		0.060	* 0.020	0.193	0.760
Physical neglect	Correlation (r) *p*	0.042	0.016	−0.045	0.148
		0.621	0.852	0.593	0.080
**LEC-5**					
Traumatic load	Correlation (r) *P*	0.185	0.160	0.068	0.221
		* 0.028	0.058	0.418	** 0.008
Accidental/injury traumas	Correlation (r) *p*	0.048	0.026	−0.013	0.118
		0.569	0.760	0.877	0.162
Victimization traumas	Correlation (r) *p*	0.242	0.243	0.100	0.212
		** 0.004	** 0.004	0.236	** 0.011
Predominant death threat traumas	Correlation (r) *p*	0.212	0.177	0.096	0.247
		* 0.011	* 0.035	0.256	** 0.003

CTQ-SF: Childhood Trauma Questionnaire–Short Form; JSPE-VP: Portuguese version of Jefferson Scale of Physician Empathy; LEC-5: Life Events Checklist for DSM-5; *n*: number; * *p* < 0.05; ** *p* < 0.01.

## Data Availability

The datasets used and analyzed during the current study are available from the corresponding author on a reasonable request.

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
