# Peer review of "General Practitioners’ Own Traumatic Experiences and Their Skills in Addressing Patients’ Past History of Adversities: A Cross-Sectional Study in Portugal"

_healthcare, 2023, doi:10.3390/healthcare11172450_

Round 1

Reviewer 1 Report

Dear authors,

Thanks for the submission. This study revealed that childhood adversities or TEs experienced of GPs have positive correlations with scores in certain dimensions of the adapted Self-confidence Scale and JSPE-VP, these correlations might help to provide better communication and understanding between GPs and their patients which is beneficial for the therapeutic process. And I agree that in general, efforts should be made at the organizational level of primary care services to promote screening for traumatic experiences.

However, there is a major concern about this study due to the larger cohort of females than male GPs, whether the differences between genders are more impactable than the childhood adversities are uncertain. It will be better to plot these data by gender as well.

Also, this study was based on the GPs and patients only in Portugal, it would be more accurate to address this in the article title. 

Reviewer 2 Report

Dear Editor,

I hope this letter finds you well. I have completed my review of the manuscript with the ID healthcare-2559066 entitled "Addressing patients’ past history of adversities in general practice: the influence of physicians’ own traumatic experiences" that you entrusted to me for evaluation.

The authors have undertaken a commendable effort in addressing an essential aspect of the GP-patient relationship concerning traumatic experiences. The manuscript is well-structured and provides an exhaustive analysis of the topic under study. The discussion section in particular draws upon pertinent literature, effectively comparing the study's results with previous research, and the authors are forthright about the strengths and limitations of their work.

While the manuscript holds significant value in its current form, there are specific areas that could benefit from further elaboration and clarity. My main suggestions revolve around:

·      Offering potential solutions or interventions to the barriers GPs face.

·      Delving deeper into the implications of the findings, particularly concerning medical education and continuous professional development.

·      Providing more context on weak statistical correlations in terms of their clinical relevance.

·      Discussing the broader global implications of their findings, especially since trauma is a universal concern.

·      Suggesting concrete strategies for primary care implementation, emphasizing the importance of trauma-informed care.

Detailed comments and recommendations are being provided below. I believe that with some refinements, this manuscript can contribute significantly to the literature in the field and provide valuable insights for GPs, healthcare institutions, and policymakers.

I would like to thank you for providing me with the opportunity to review this manuscript. It has been an enriching experience, and I hope my feedback will assist the authors in enhancing their valuable work.

Please do not hesitate to contact me should you require any further clarification or information regarding my review.

Warm regards,

ABSTRACT

The abstract presents a potentially valuable study that explores the often-overlooked personal experiences of GPs and how they might impact their professional skills and patient interactions. With some refinements for clarity and more concrete data presentation, this abstract can effectively represent the depth and significance of the research.

1.     The initial statement "Addressing trauma has been found to be important for primary care patients" could be strengthened by specifying why it is essential.

2.     The objective is clear, but it might benefit from a brief elaboration on why understanding this influence is essential for clinical practice or patient outcomes.

3.     It would be helpful to clarify whether the survey's dissemination via email and social media groups was random or if specific criteria were followed.

4.     The participation of 143 GPs is noted, but it might be constructive to provide some context: was this a significant percentage of the targeted group, or does it represent a small fraction of the total GPs in Portugal?

5.     It would be beneficial to provide more quantitative data, e.g., percentage of GPs identifying specific barriers, or mean scores for self-confidence and empathy.

6.     While the correlation of GPs' exposure to TEs and ACEs with certain dimensions of self-confidence and empathy is mentioned, it would be beneficial to specify which dimensions showed significant correlations.

7.     The correlation coefficients (e.g., r-values) might add to the quantification of the relationships if space allows.

8.     The phrase "GPs identified barriers to routinely address trauma, such as lack of time, fear of causing more suffering to the patient, among others." might be better phrased as "GPs identified barriers to addressing trauma routinely, including a lack of time and a fear of causing further patient suffering."

9.     The conclusion presents an interesting paradox: while GPs with traumatic experiences feel more confident and empathetic, they still identify barriers like lack of training and proper care conditions. This point could be slightly expanded upon to provide clarity. A recommendation or implication based on this paradox might add value to the abstract.

10.  The authors may consider clarifying or expanding upon the "better patient care conditions" in the last sentence. It's not immediately clear what is meant by this phrase, and specifics could help in providing a clearer picture.

11.  Given the findings, it might be beneficial to include a concise sentence or two on potential recommendations or implications for clinical practice in primary care settings.

INTRODUCTION

The introduction section is well-composed and articulates the significance of the subject concerning adverse childhood experiences (ACEs) and traumatic events (TEs), particularly within the context of Portuguese General Practitioners (GPs). The authors succeed in setting the scene, providing relevant background, and outlining the research's primary aim and hypotheses. However, there are areas where the text can be improved to enhance clarity, precision, and academic rigor.

1.     Lines 34-35: The definition of Adverse childhood experiences (ACEs) could benefit from some refinement for clarity. Consider: "Adverse childhood experiences (ACEs) refer to direct or indirect harm caused to children, primarily resulting from abuse, neglect, and tumultuous domestic environments."

2.     Line 44-46: The inclusion of specific statistics related to ACEs in Portugal enriches the context but may benefit from comparison with other countries or broader contexts to illustrate its significance.

3.     Line 48-49: Consider rephrasing the phrase "they can disrupt, both directly and indirectly," to "they can directly and indirectly disrupt."

4.     Lines 53-62: The introduction of Traumatic Events (TEs) feels somewhat abrupt. It would be helpful to provide a transition between ACEs and TEs to guide the reader's understanding of their interconnectedness or differences.

5.     Line 67: There appears to be a typographical error in the word "approaching."

6.     Line 69-70: Elaborate a bit more on the term "existential dimensions." Not all readers might be familiar with its application in this context.

7.     Lines 75-77: Consider rephrasing for clarity: "Existing research shows limited focus on GPs' perspective on the importance of addressing past traumatic experiences. Moreover, the methods GPs employ to communicate about these experiences in clinical practice remain underexplored."

8.     Line 80: It might be helpful to mention whether the "breaking of the doctor-patient relationship" refers to a perceived risk or documented occurrences.

9.     Lines 80-81: The statement "both ACEs and TEs are neglected topics when it comes to primary health care" is quite strong. It would be good to provide a bit more context or evidence for this, even if it's further elaborated upon in the manuscript's later sections.

10.  Line 84: The word "inumerous" seems to be a typographical error and should be corrected to "numerous."

11.  Line 86: The spelling of "likely" appears incorrect as "likley."

12.  Lines 82-89: The transition from discussing barriers for GPs to discussing the GP's own history with ACEs and TEs is a little abrupt. Consider introducing this shift with a statement on the potential influence of personal histories on professional practices. Additional context or theoretical framework may enhance understanding.

13.  Lines 93-103: It would be useful to have a brief sentence or two setting the global or regional context of this study. For instance, how prevalent are ACEs and TEs among Portuguese GPs compared to other nations or regions?

14.  Lines 98-102: The hypotheses formulated are clear but may benefit from a more detailed explanation or rationale, connecting them to existing literature or theoretical underpinnings.

15.  The introduction might benefit from a brief summarizing paragraph that succinctly restates the research problem and purpose, linking it to the broader significance in healthcare practice.

16.  The authors may add a sentence or two highlighting the broader significance of the study in the global context.

17.  The authors may consider adding a sentence about the implications of the study. Why is understanding GP's perceptions and personal histories crucial in the larger context of primary healthcare?

MATERIALS AND METHODS

The Materials and Methods section of the manuscript provides a comprehensive description of the participant's recruitment, study instruments, and statistical analysis. However, there are some areas that need attention and clarification to enhance the quality and reproducibility of the study. Here are my specific comments and suggestions:

1.     More information should be provided about the recruitment process, including the criteria for selection and exclusion. Did the authors include all the GPs in Portugal, or were there specific qualifications or criteria for participation? The method of participant selection might introduce selection bias, especially through social media groups and GPs' associations. How were these sources chosen? Was there any criterion for the selection of these sources?

2.     Specify the total number of GP specialists and residents to whom the survey was sent to determine the response rate.

3.     The authors mentioned using Google Forms® for survey creation but did not provide details about the survey’s validation, such as pre-testing or piloting. More information about how the survey questions were developed and validated would enhance the robustness of the method.

4.     It is good to see that ethical approval was obtained. However, information about the data anonymization and the measures taken to ensure participants' privacy and confidentiality should be included to conform to ethical standards. For instance, using Google Forms for collecting confidential data may raise ethical concerns, especially regarding the security and anonymity of the data. Clarify the measures taken to ensure data privacy.

5.     The authors should provide rationale or statistical justification for the chosen sample size to ensure that it provides sufficient power for the analyses.

6.     The specific content of the survey (other than the standardized scales) is not presented. Consider briefly describing the content or types of questions included, especially in the fourth section.

7.     Ensure that the unpublished source "Ferreira, Ribeiro, Santos & Maia, 2016" is correctly cited and formatted according to the journal's guidelines. If this is a personal communication, it may not be suitable as a citation.

8.     Please clarify whether the Portuguese version of the CTQ-SF was validated and previously used in the population under study.

9.     The rationale behind not considering the analysis of item 17 in the LEC-5 needs to be better justified.

10.  When mentioning "two surveys were used to evaluate GPs' clinical communication skills," only one (Self-confidence Scale for clinical communication skills) is described. The second one should either be detailed or the sentence should be corrected.

11.  The authors mentioned the adaptation of the Self-confidence Scale. More information about how the adaptation was done and whether the adapted scale was validated would be essential.

12.  While the authors have mentioned the value of Cronbach's alpha for several scales, a brief discussion or reference to the acceptable thresholds for reliability could add context to these values.

13.  The statistical analysis section appears to be well described, but the authors should specify if any adjustments were made for multiple comparisons and the methods used to handle missing data, if any.

14.  Although the Kolmogorov-Smirnov test was used to check normality, it would be good to mention any other assumptions checked, and if any transformations or non-parametric tests were considered for non-normally distributed data.

15.  Clarification on how assumptions for the parametric tests were met or what actions were taken if they were violated would provide more rigor to the statistical analysis.

16.  Exploratory factor analysis is mentioned for the adapted Self-confidence Scale, but no extraction method (e.g., principal component analysis) or rotation method (e.g., varimax) is specified. Please provide this information. 

17.  Specify the criteria for factor loading cut-offs when discussing the exploratory factor analysis.

18.  It might be useful to include the citation or reference for the SPSS® software used in the analysis.

RESULTS

The results section of the manuscript provides a detailed account of the study's outcomes and offers insights into GPs’ clinical communication skills and their associations with years of experience, previous training, and past traumatic experiences. However, there are areas where the clarity, organization, and presentation can be improved to enhance reader understanding. 

1.     There appears to be a typo in section 3.1, line 214: "communiczation" should be corrected to "communication."

2.     Line 215: "With regards GPs’ clinical communication perceptions" might be better written as "Regarding GPs’ clinical communication perceptions". 

3.     Section 3.2 discusses significant associations but also mentions non-significant results. It might be helpful to elaborate on why these non-significant results are important for the study or to frame them within the broader context of the research.

4.     Line 251: Change "Accordingly it can bee seen" to "Accordingly, it can be seen".

5.     The authors should ensure consistency in reporting statistical results. For instance, they may consider including effect sizes where appropriate to supplement the significance values and provide a measure of the practical significance of the findings.

6.     The authors should ensure that the scales' directions are clear. For example, is a higher score on the JSPE-VP indicative of more or less empathy? This context can help readers better interpret the results.

7.     While the p-values are mentioned, the authors may consider also reporting the exact F or r values in the text (where not already provided) for completeness. 

8.     Given the detailed data on GPs' views on addressing trauma in patients, it might be insightful to explore any potential gender differences or other demographic variables that might influence these perspectives.

DISCUSSION

The manuscript provides insightful observations about GPs’ approach to patients’ traumatic experiences, reasons why patients may or may not discuss traumas, and various factors that influence GP's confidence and empathy. However, several areas would benefit from further clarification, elaboration, or adjustment.

1.     The section starts directly with the results without setting the stage for the reader. Consider starting with a brief summary of the purpose of the study to help orientate the reader. 

2.     When mentioning prior studies ([19, 20, 31, 32]), it would be helpful to briefly describe the specifics of these studies to contextualize your results and further emphasize their significance. Analyzing and contrasting with other similar works would offer a more comprehensive understanding of how this study fits within the existing literature.

3.     While the reasons cited by GPs for not asking about traumatic experiences are clear and well-outlined, it would be helpful if the authors could comment on any potential strategies to mitigate these barriers. For instance, training opportunities or resources that might be introduced to assist GPs. 

4.     The authors highlight a potential gap in communication or understanding between GPs and patients. It may be worthwhile to provide recommendations for GPs on how they might bridge this gap. How can GPs become more attuned to these "small hints" from patients? 

5.     It's interesting to note the positive association between clinical experience and GP confidence. This might be an area to delve into further in future research. What other factors might boost GP confidence?

6.     The section about the association between empathy and clinical experience could benefit from a more in-depth exploration of why these two factors may not be linked, especially since the idea of decreasing empathy over time is quite intriguing. 

7.     The mention of gender differences (76.9% of the participants being female) and its potential effect on the results is significant. The authors should discuss this aspect more thoroughly and explain why gender was not adjusted in the analysis, especially considering the evidence cited regarding differences in empathy scores.

8.     The range of correlation values mentioned (0.010 to 0.237) indicates weak associations. This warrants a discussion on whether these correlations are practically significant and how they should be interpreted in the context of the study.

9.     It's commendable that the study's limitations, such as the lack of adjustments for age or gender and the constraints of an online survey, are addressed. However, it might be helpful to also explore how these limitations specifically might have affected the present results. Furthermore, the proposed future research directions could be expanded to include specific methodologies or populations that should be explored.

10.  The limitations associated with the online survey are appropriately acknowledged. However, given the ever-increasing digitization of research, it might be worthwhile to also mention any potential benefits or strengths of online surveys in addition to their limitations.

11.  When discussing the idea of "post-traumatic growth" and how experiencing trauma might make GPs more empathetic, it would be beneficial to provide more insight into how this growth typically manifests and its general implications. 

12.  The final paragraph briefly touches on implications for practice, like changes at the organizational level. This could be further expanded upon to provide a clearer roadmap for future practices. How exactly can GPs’ medical education emphasize the importance of addressing traumatic experiences? What specific organizational changes could be implemented in primary care services? 

13.  It may be useful to include potential interventions or training programs that can be developed based on this study's findings to improve the GP-patient relationship in the context of traumatic experiences. 

14.  Given the importance of the topic, it might be helpful to touch upon potential policy implications and recommendations for healthcare institutions. 

15.  Consider discussing the potential global relevance of your findings. While the study is based in Portugal, trauma and its implications are a universal concern.

There are a few typos that need correction.

Reviewer 3 Report

Thank you for the opportunity to review this interesting manuscript. I have a few minor comments for the authors to consider:

·        Title:

o   The title does not reflect the aim of the study. Perhaps the authors could revise it to “Physicians’ own traumatic experiences and their skill addressing patients’ past history of adversities: a cross-sectional study”

·        Abstract:

o   Line 20: Please change the word “assessed with” to “assessed using”

o   Line 23-25: Please include the correlation coefficient (r).

o   Line 25-27: Please include the beta coefficient (or adjusted beta coefficient) and its 95% CI to illustrate the magnitude of the difference in confidence.

·        Methods:

o   Line 106: Please clarify if ALL GP specialists and residents were approached. If not, please include the sampling frame, e.g. GP specialists and residents specialising in Traumatic Events in the metropolitan area.

o   Line 110: Please remove the redundant word “observational” from the phrase “In this cross-sectional observational study” because it is known that cross-sectional study is an observational study.

o   Line 123: The authors stated that “The questionnaire of the survey was specifically designed for this study, consisting of four sections.” Please outline the process the authors did to assess the validity and reliability of the questionnaire, e.g. face validity for Section 1; content and construct validity for Section 2, test-retest reliability for Section 4, etc. If validity/reliability check was not performed, please acknowledge this in the Discussion (limitation) section.

o   Line 188: Please revise “The value of 5% was adopted …” to “The alpha level of 5% was adopted…”

o   Line 189-190: Instead of stating the use of ANOVA, please clearly relate it to the research hypotheses. [Note that ANOVA may not be the most appropriate test, depending on the specific research hypotheses.] Eg. Score of difficulty in routinely approaching patients by GPs with and without previous encounter was assessed using independent-sample t-test. Please also outline the steps to assess if assumptions of the statistical tests were violated.

o   Line 191: This statement on normality assessment should be mentioned upfront, after the first sentence on line 183.

o   Line 193-194: This statement should be placed on line 189 after the alpha level of 5%.

o   It would be good to perform multiple linear regression to illustrate the effect of the various measures together on the outcome of interest. Please justify why multiple linear regression (MLR) is not performed, especially when gender or age could also be adjusted in the MLR to resolve the study limitations mentioned by the authors.

·        Conclusions:

o   Line 380: ‘hampered by’ is not a suitable word. Suggest to revise to ‘affected by’.

Minor editing would be appreciated.
